# Multivessel Coronary Artery Dissection in a Patient with Co-Occurrence of Aortic Dissection and Dilated Cardiomyopathy in the Postpartum Period

**DOI:** 10.3390/diseases11040178

**Published:** 2023-12-10

**Authors:** Takahiro Kurihara, Eisuke Amiya, Masaru Hatano, Junichi Ishida, Shun Minatsuki, Shunsuke Inoue, Seitaro Nomura, Hiroyuki Morita, Issei Komuro

**Affiliations:** 1Department of Cardiovascular Medicine, Graduate School of Medicine, The University of Tokyo, Bunkyo-ku, Tokyo 113-8655, Japan; 2Department of Therapeutic Strategy for Heart Failure, Graduate School of Medicine, The University of Tokyo, Bunkyo-ku, Tokyo 113-8655, Japan; 3Department of Advanced Medical Center for Heart Failure, Graduate School of Medicine, The University of Tokyo, Bunkyo-ku, Tokyo 113-8655, Japan; 4Department of Frontier Cardiovascular Science, Graduate School of Medicine, The University of Tokyo, Bunkyo-ku, Tokyo 113-8655, Japan; 5Department of Cardiovascular Medicine, Graduate School of Medicine, International University of Health and Welfare, Minato-ku, Tokyo 107-8402, Japan

**Keywords:** coronary artery dissection, dilated cardiomyopathy, aortic dissection, postpartum

## Abstract

The co-occurrence of dilated cardiomyopathy (DCM) and aortic dissection has been rarely reported. Here, we present the case of a patient with co-occurrence of DCM and aortic dissection, wherein multivessel coronary artery dissection eventually occurred, thereby leading to advanced heart failure. She suffered from co-occurrence of DCM and aortic dissection 6 years ago. After the heart failure had briefly stabilized, the myocardial infarction due to coronary artery dissection led to worsening mitral regurgitation and decreased right ventricular function, thereby worsening the status of her heart failure. In addition to cardiovascular abnormalities, the patient was also complicated by short stature (145 cm), mild scoliosis, nonfunctioning pituitary adenoma of 1 cm in size, and retinitis pigmentosa. Coronary artery dissection is a possible complication in patients with co-occurrence of DCM and aortopathy, which could dramatically affect the clinical course of advanced heart failure.

## 1. Introduction

The co-occurrence of dilated cardiomyopathy (DCM) and aortic dissection has been rarely reported, and it has not been concisely described. Here, we present the case of a patient with co-occurrence of DCM and aortic dissection, wherein multivessel coronary artery dissection eventually occurred, thereby leading to advanced heart failure.

## 2. Case Presentation

A 30-year-old woman experienced the first episode of acute heart failure 10 days after vaginally delivering her first child. On echocardiography, the left ventricular ejection fraction (LVEF) was mildly reduced (47%) without overt dilation (left ventricular diastolic diameter (LVDd) 50 mm). Due to a lack of any other identifiable cause, postpartum cardiomyopathy was suspected [1], and the patient received medications, including furosemide, carvedilol, losartan, and spironolactone; however, her B-type natriuretic peptide (BNP) level gradually increased, and she was hospitalized for heart failure (Figure 1A). Following hospitalization, dobutamine support with diuretic adjustment gradually improved her status; however, the patient developed hoarseness and back pain at 14 days of hospitalization (Figure 1B). Contrast-enhanced computed tomography (CT) revealed a Stanford B-type aortic dissection that extended from the distal aortic arch to the descending aorta of the lower renal artery bifurcation without thrombus formation (Figure 2A,B). Following aortic dissection onset, heart failure further worsened, and the patient was considered dobutamine-dependent; therefore, she was transferred to our hospital.

At the time of transfer, the patient’s BNP level was 450 pg/mL, and echocardiography showed a significant decrease in cardiac function (LVEF, 24%; LVDd, 47 mm) (Figure 2D). Coronary CT revealed no coronary artery stenosis detected. Right heart catheterization showed a mean right atrial pressure (mRAP) of 6 mmHg, mean pulmonary artery pressure (mPAP) of 31 mmHg, mean pulmonary artery wedge pressure (mPAWP) of 18 mmHg, and cardiac index (CI) of 1.88 L/min/m^2^. Myocardial biopsy showed no inflammatory cell infiltrates and only mild fibrosis. After the transfer, heart failure was alleviated by optimizing medical treatment, whereas back pain due to DA was decreased and no worsening of heart failure was observed, which led to the successful tapering of dobutamine. After adjusting medication doses, including an increase in carvedilol dosage, the BNP level improved to approximately 200 pg/mL, and the patient was discharged 3 months later. Her electrocardiogram showed a narrow QRS complex, for which cardiac resynchronization therapy was not recommended. In addition, there had been no episode of non-sustained ventricular tachycardia in electrocardiogram and Holter monitoring. During hospitalization, we screened for collagen disease; however, autoantibody and antineutrophil cytoplasm autoantibodies examination showed no abnormalities. We examined the thyroid, suprarenal gland, and sex hormones, and the result showed the following values: adrenocorticotropic hormone, 8.4 pg/mL; cortisol, 15.0 μg/dL; thyroid-stimulating hormone, 3.49 μIU/mL; follicle-stimulating hormone, 14.1 mIU/mL; luteinizing hormone, 2.9 mIU/mL; prolactin, 8 ng/mL; estradiol, 17.0 pg/mL; growth hormone (GH), 0.08 ng/mL; insulin-like growth factor-I, 94 ng/mL; and aldosterone, 282 pg/mL. All of these values, but GH, were within the normal range, whereas the level of GH was slightly lower than the normal range. The patient had a family history of DCM (her mother had heart failure due to DCM). She had no siblings. In addition to cardiovascular abnormalities, the patient was also complicated by short stature (145 cm), mild scoliosis, nonfunctioning pituitary adenoma of 1 cm in size, and retinitis pigmentosa. Based on these multiple abnormalities, including young-onset aortic dissection and cardiomyopathy, we considered a hereditary disease of the myocardial and aortopathy phenotype; however, no specific genetic abnormalities in known aortopathy-related genes (*ACTA2*, *TGFBR2*, *SMAD3*, *FBN1*, and *COL3A1*) could be detected, nor any known cardiomyopathy-related genes, which were screened using whole exome sequencing. The patient did not have a history of alcohol consumption, morbid obesity, biopsy/imaging-confirmed myocarditis, or cardiotoxic chemotherapy, which are the risk factors for DCM development [2]. Regarding the family history of extracardiac findings, her mother and the two sisters of her mother had short statures. One sister of her mother had diabetes, whereas the other two sisters did not have cardiac abnormalities.

Following discharge, she continued to receive medical treatment at an outpatient clinic, and her heart failure was kept stabilized for approximately 6 years. Despite the maximum tolerable dosing of heart failure medications, her heart did not achieve reverse remodeling, and her cardiac function remained severely impaired. Sacubitril/valsartan and sodium–glucose cotransporter 2 inhibitors were not available at that time. Owing to severely impaired cardiac function, she was not recommended to have any subsequent pregnancies.

At 37 years old, the distal arch aortic aneurysm had expanded to 50 mm (Figure 2C), and distal arch replacement was performed (Figure 1C). The postoperative course was uneventful; she was discharged without any signs of heart failure worsening. Eight months following discharge, the patient developed right-sided abdominal pain and mild dyspnea and was readmitted to the hospital. Coronary angiography revealed an acute inferior wall infarction due to coronary artery dissection at the right coronary artery (RCA) #3, and the patient underwent percutaneous coronary stenting at this site (Figure 1D and Figure 2E,F). Peak creatine kinase levels were approximately 1000 U/L. On echocardiography, the LVEF was 28%, unchanged from the previous level. Her postoperative course was uneventful, and she was discharged 2 weeks later. However, following discharge, she continued to experience chest discomfort during exertion at home, and her BNP level increased to 700 pg/mL. The patient was readmitted to the hospital within a few months. On admission, the BNP level increased to 699 pg/mL; echocardiography revealed LVEF of 20%, which was not changed before heart failure worsened. The exacerbation of moderate mitral regurgitation (MR) was newly detected (Figure 2H), which may be functional MR exacerbation due to the worsening of ischemia. Furthermore, the right ventricular fraction area change (RVFAC), which had been consistently maintained, markedly decreased, indicating a significant decline in right heart function due to ischemia (Figure 1). A coronary CT revealed the progression of the area of coronary artery dissection, including the proximal portion of the stent in the RCA #1–#2 (Figure 2I) and left circumflex #13 (Figure 2J). These ischemia could explain the exacerbation of MR and the decline in RV function. Moreover, the LAD, wherein coronary artery dissection was not observed, was found to have diffuse aneurysmal dilation compared with the previous one (Figure 2K,L). Right heart catheterization showed low output (CI, 2.18 L/min/m^2^) with normal intracardiac pressures (mRAP, 4 mmHg; mPAP, 16 mmHg; mPAWP, 13 mmHg). However, the right ventricular stroke work index (RVSWI) was markedly decreased (10.6 → 4.9 g·m/m^2^/beat). Treating coronary artery dissection occurring simultaneously over multiple vessels was challenging and not sufficiently beneficial. After some medication adjustments, she was discharged from the hospital 1 month later. However, her heart failure status could not subsequently be stabilized. The patient required repeated hospitalization for heart failure treatment. For this medically intractable heart failure, we proposed an implantation of a left ventricular assist device. However, the patient disagreed because it was difficult for her to gain family support for the treatment. At 38 years old, 11 months following the coronary artery dissection, the patient died of heart failure.

## 3. Discussion

In this case report, a patient with DCM concurrent with aortic dissection finally developed multivessel coronary artery dissection and reexacerbated heart failure due to myocardial ischemia. The instructive point is that the coronary artery can also be involved as a lesion in DCM with aortic dissection. In addition, the vascular events, including aortic dissection and coronary artery dissection, significantly affected the clinical course of advanced heart failure. To the best of our knowledge, this is the first case report about the co-occurrence of aortic dissection, DCM, and coronary artery dissection.

First, we reviewed cases in which DCM is complicated by aortopathy. There were some reports about cases wherein co-occurring aortic dissection and DCM were noted. The presence of family history suggested that the etiology of decreased myocardial function is derived from genetic abnormalities. However, in this case, her mother with DCM had no findings of aortopathy. Additionally, other abnormalities, including short stature, mild scoliosis, nonfunctioning pituitary adenoma, and retinitis pigmentosa, were observed, suggesting that the patient’s genetic predisposition was different from her mother’s. Some similar reports demonstrated aortic dissection and postpartum cardiomyopathy in a young postpartum woman [3]. For instance, Marfan and Loeys–Dietz syndromes have been reported to be associated with cardiac dysfunction [4], wherein aortopathies are more likely to develop in the postpartum period. However, we could not confirm these diagnoses on the basis of clinical and genetic findings. A characteristic finding in this case was that aortic dissection occurred with heart failure during the postpartum period, and heart failure was significantly aggravated by the occurrence of aortic dissection.

Regarding coronary artery dissection, some inherited disorders have been reported to be complicated by spontaneous coronary artery dissection (SCAD). Several studies have reported on the genetic causes of SCAD [5,6], and SCAD shares genetic overlap with connective tissue disease. A recent study has reported that 3.5% of the patients with SCAD had causal or likely pathogenic rare genetic variants, mostly in genes associated with other known disorders (e.g., vascular Ehlers–Danlos syndrome, Loeys–Dietz syndrome, and adult polycystic kidney disease) [7]. Inherited vasculopathies, including Marfan syndrome, Loeys–Dietz syndrome, vascular Ehlers–Danlos syndrome, α1-antitrypsin deficiency, and polycystic kidney disease, may be complicated with SCAD [8]. However, in the present case, characteristics of these etiologies, and possible genetic variants were not identified in the present case. Furthermore, in this case, diffuse coronary artery dilation was observed, and it seems reasonable to consider that coronary artery dissection was the result of a pathological condition similar to coronary artery dilatation.

In this case, following coronary artery dissection development, heart failure became uncontrollable and eventually became difficult to treat (Figure 3). This coronary artery dissection was the second event wherein vascular dissection significantly worsened the clinical course of heart failure. Of course, the main cause of worsening heart failure was severe systolic dysfunction, but it is expected that additional other factors may have added to the worsening of heart failure. Ischemia of the RCA was reported to correspond to right ventricular dysfunction [9]. Additionally, inferior myocardial infarction contributed to ischemic MR development. Although it is a rare disease, it was suggested that considering the possibility of coronary artery dissection in patients with DCM and aortic dissection is necessary. As coronary artery dissection may be a possible outcome in patients experiencing DCM and aortic dissection, treatments, including heart transplantation, should be considered at the earlier stage. Coronary artery disease is caused by various factors, and among these, the contribution of immune response has recently been attracting attention [10]. Although immune reaction may have contributed to the coronary events in the current case, this is not fully established, and future research is desired.

Although systolic blood pressure during outpatient follow-up was continuously maintained below 80 mmHg, aortic and coronary artery dilation could not be prevented. Careful monitoring of the artery diameter at multiple sites may lead to the appropriate decision in the therapeutic strategy, including heart failure.

As a limitation, it is believed that the various abnormalities exhibited by the patient are related to genetic factors; however, those factors could not be determined. Although the GH level was low, comprehensive evaluations of hormones were not performed. There were some studies investigating the link between low GH and aortic dilation. For instance, Feingold 2 syndrome has characteristics such as GH deficiency associated with adenohypophyseal compression, aortic dilation, phalangeal joint contractures, memory and sleep problems in addition to the typical features of microcephaly, brachymesophalangy, toe syndactyly, short stature, and cardiac anomalies [11]. Additionally, the link has also been investigated in cases of Turner syndrome. Turner syndrome is defined as a complete or partial absence of one sex chromosome in a phenotypic female, which is characterized by short stature and ovarian failure, and the most medically significant feature is congenital heart disease with a high risk for aortic dilation and dissection. Bondy C et al. investigated the effect of GH on aortic dilation, which suggested the GH treatment of girls with Turner syndrome does not seem to affect the ascending or descending aortic diameter above the increase related to the larger body size [12]. There is still room for investigation into the relationship between GH and aortic dilation. Furthermore, screening for genetic abnormalities in family members was inadequately performed. Regarding genetic abnormalities, epigenetic mechanisms are also being discussed in the field of a wide range of cardiovascular diseases [13]. One recent study has reported that epigenetic regulation is also involved in aortic dissection [14]. Epigenetic factors may also be involved in the present; however, this was insufficiently considered.

## 4. Conclusions

Coronary artery dissection is one possible complication in patients with the co-occurrence of DCM and aortopathy, which could dramatically affect the clinical course of advanced heart failure.

## Figures and Tables

**Figure 1 diseases-11-00178-f001:**
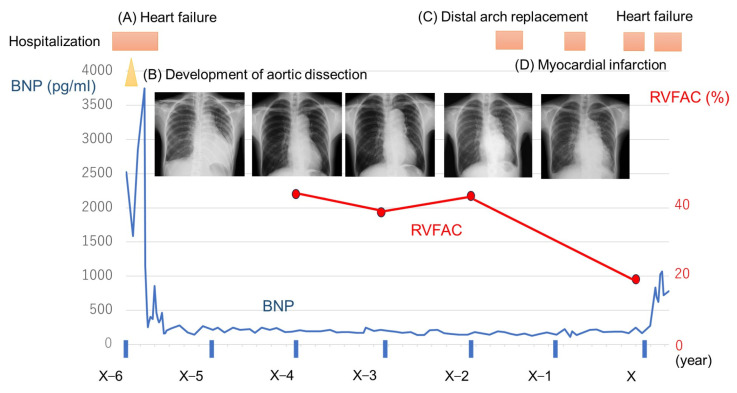
Time course of B-type natriuretic peptide (BNP) and findings of chest X-rays and plots of right ventricular fraction area change (RVFAC). (**A**) First episode of hospitalization due to heart failure. (**B**) During hospitalization for heart failure, aortic dissection occurred, which further worsened heart failure. (**C**) Distal arch replacement for the dilated aorta due to aortic dissection. (**D**) Acute myocardial infarction due to coronary artery dissection.

**Figure 2 diseases-11-00178-f002:**
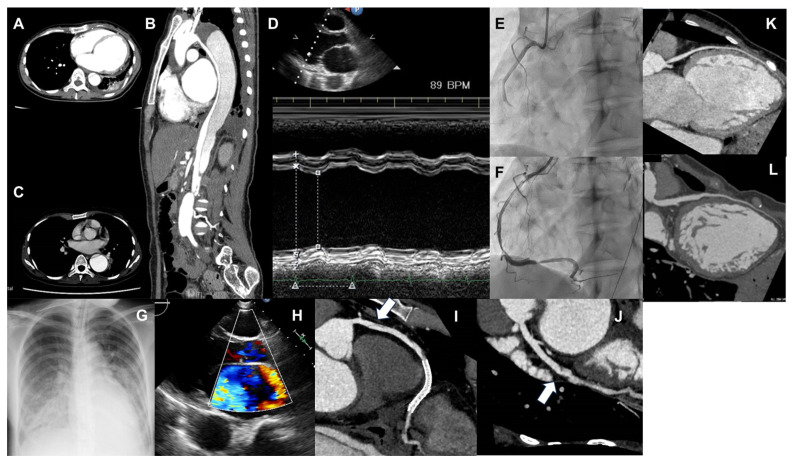
(**A**,**B**) Computed tomography (CT) image showing aortic dissection. A is the horizontal section, and B is the vertical section. A false lumen is observed from the distal arch, which seems to compress the true lumen. (**C**) CT image of dilated aortic dissection at the distal arch, which has expanded to 50 mm, and it has been determined that distal arch replacement surgery should be performed. (**D**) Image of M-mode in echocardiography, showing a markedly reduced left ventricular contractility. (**E**,**F**) Coronary angiography images before and after right coronary revascularization. (**E**) Showing the right coronary artery is occluded at #3 due to coronary artery dissection. (**F**) A coronary stent being placed in #3. (**G**) Chest radiograph showing pulmonary edema due to heart failure exacerbation. (**H**) Mitral regurgitation exacerbation due to myocardial ischemia. (**I**) Image of coronary CT angiography of the right coronary artery. Widespread coronary artery dissection on the proximal portion (#1–#2) of the right coronary artery stent. (**J**) Image of coronary CT angiography of the left circumflex artery. Coronary artery dissection is newly recognized in the left circumflex (#13). (**K**) Image of coronary CT angiography of the left ascending artery in the first hospitalization for heart failure. (**L**) Image of coronary CT angiography of the left ascending artery following the development of inferior myocardial infarction. The left ascending artery is diffusely dilated as compared with that observed in Figure (**K**).

**Figure 3 diseases-11-00178-f003:**
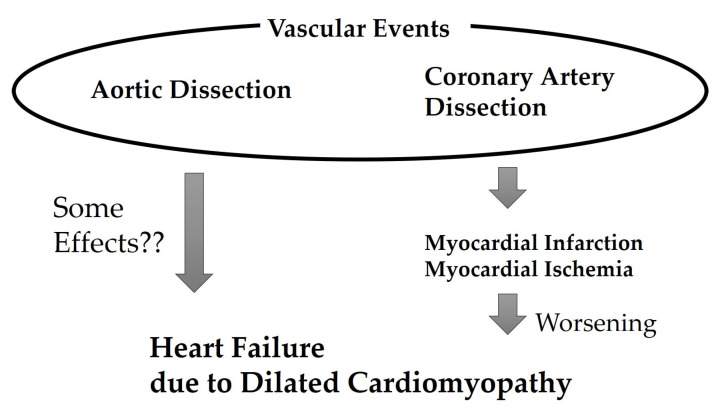
Association between heart failure worsening and vascular events in the present case.

## Data Availability

Data are available upon request.

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
