# Peer review of "Multivessel Coronary Artery Dissection in a Patient with Co-Occurrence of Aortic Dissection and Dilated Cardiomyopathy in the Postpartum Period"

_diseases, 2023, doi:10.3390/diseases11040178_

Round 1
Reviewer 1 Report
Comments and Suggestions for Authors
The authors report a case of a patient with postpartum cardiomyopathy in which the simultaneous occurrence of coronary artery dissection aggravated heart failure. The patient's family background and the comorbid complications also provide meaningful information in elucidating causative factors. This case report offers valuable information that aortopathy promotes advanced heart failure.
The reviewer has some comments, as indicated below.
[Major comments]
1. Is it conceivable that this patient and her mother may have had common risk factors for DCM? The information regarding other siblings would be helpful.
2. If possible, please comment on the risk factors, including epigenetic mechanisms, that led to the development of SCAD and coronary artery dilatation in the postpartum period in this patient in the Discussion section.
[Minor comment]
P. 4 line 119: The abbreviation should be shown in the parenthesis, such as mitral regurgitation (MR). Hereafter, mitral regurgitation can be abbreviated as MR.
Author Response
Dear Editor and Reviewers
We thank the referee for fruitful suggestions. We have revised our manuscript on the basis of the referee’s comments. We look forward to a publication of our manuscript in your journal. Our responses to the referee’s comments are as follows:
The authors report a case of a patient with postpartum cardiomyopathy in which the simultaneous occurrence of coronary artery dissection aggravated heart failure. The patient's family background and the comorbid complications also provide meaningful information in elucidating causative factors. This case report offers valuable information that aortopathy promotes advanced heart failure.
Response
Thank the reviewer for the pertinent comment.
[Major comments]
- Is it conceivable that this patient and her mother may have had common risk factors for DCM? The information regarding other siblings would be helpful.
Response
Thank the reviewer for the pertinent comment. According to risk factors for the development of DCM, several factors might affect it in patients with genetic susceptibility to cardiomyopathy. # They included morbid obesity, high amount of alcohol consumption, cardiotoxic chemotherapy, biopsy/imaging-confirmed myocarditis and perpartum. We added a comment about it in revised manuscript. We also added the information about family history of extracardiac findings such as short statue.
# Giudicessi JR, Shrivastava S, Ackerman MJ, Pereira NL. Clinical Impact of Secondary Risk Factors in TTN-Mediated Dilated Cardiomyopathy. Circ Genom Precis Med. 2021 Apr;14(2):e003240. doi: 10.1161/CIRCGEN.120.003240. Epub 2021 Apr 19. PMID: 33866824; PMCID: PMC8284363.
Change
She did not have history of high amount of alcohol consumption, morbid obesity, biopsy/imaging-confirmed myocarditis and cardiotoxic chemotherapy, which are the risk factors of the development of DCM.
In terms of family history of extracardiac findings, her mother and two mother’s sisters had short statue. One mother’s sister had diabetes, however two mother’s sisters did not have cardiac abnormalities.
- If possible, please comment on the risk factors, including epigenetic mechanisms, that led to the development of SCAD and coronary artery dilatation in the postpartum period in this patient in the Discussion section.
Response
Thank the reviewer for the pertinent comment. As the reviewer pointed out, we added a comment about it in discussion.
Change
Regarding genetic abnormalities, epigenetic mechanisms are also being discussed in the field of a wide range of cardiovascular diseases.#1 One recent report had shown that epigenetic regulation is also involved in aortic dissection.#2 There was a possibility that epigenetic factors were involved in this case as well, but this had not been considered sufficiently.
#1 Tao Y, Li G, Yang Y, Wang Z, Wang S, Li X, Yu T, Fu X. Epigenomics in aortic dissection: From mechanism to therapeutics. Life Sci. 2023 Nov 6;335:122249. doi: 10.1016/j.lfs.2023.122249. Epub ahead of print. PMID: 37940070.
#2 Chakraborty A, Li Y, Zhang C, Li Y, Rebello KR, Li S, Xu S, Vasquez HG, Zhang L, Luo W, Wang G, Chen K, Coselli JS, LeMaire SA, Shen YH. Epigenetic Induction of Smooth Muscle Cell Phenotypic Alterations in Aortic Aneurysms and Dissections. Circulation. 2023 Sep 19;148(12):959-977. doi: 10.1161/CIRCULATIONAHA.123.063332. Epub 2023 Aug 9. PMID: 37555319; PMCID: PMC10529114.
[Minor comment]
- 4 line 119: The abbreviation should be shown in the parenthesis, such as mitral regurgitation (MR). Hereafter, mitral regurgitation can be abbreviated as MR.
Response
Thank the reviewer for the important comment. We corrected it in revised manuscript.

Reviewer 2 Report
Comments and Suggestions for Authors
The case is intereseting, the presentation includes the most important elements. However, for me there are some questions pending:
1. cardiac MRI data related to the cardiomyopathy, or in the later phases of the disease?
2. hormone status? thyroid, suprarenal gland, sex hormones?
3. subsequent pregnancies?
4. ICD, CRT indication?
5. COVID, autoimmun diseases, lues?
6. Some minor mistakes in English expressions, e.g. coronary ischemia (ischemia is enough), etc.
Comments on the Quality of English Languageminor editing required
Author Response
Dear Editor and Reviewers
We thank the referee for fruitful suggestions. We have revised our manuscript on the basis of the referee’s comments. We look forward to a publication of our manuscript in your journal. Our responses to the referee’s comments are as follows:
Reviewer 2
The case is interesting, the presentation includes the most important elements. However, for me there are some questions pending:
- cardiac MRI data related to the cardiomyopathy, or in the later phases of the disease?
Response
Thank the reviewer for the important comment. However, she had not received cardiac MRI study. Therefore, we unfortunately cannot provide cardiac MRI results.
- hormone status? thyroid, suprarenal gland, sex hormones?
Response
Thank the reviewer for the pertinent comment. We added the data of several hormones in revised manuscript. In addition, we added a comment about the limited data of hormones in limitation section.
Change
We measured thyroid, suprarenal gland and sex hormone. It showed that adrenocorticotropic hormone; 8.4 pg/ml, cortisol; 15.0 μg/dL, thyroid-stimulating hormone; 3.49 μIU/mL, follicle-stimulating hormone; 14.1 mIU/mL, luteinizing hormone; 2.9mIU/mL, prolactin; 8 ng/mL, estradiol; 17.0 pg/mL, growth hormone; 0.08 ng/mL, insulin-like growth factor-I; 94 ng/mL, and aldosterone; 282 pg/mL. All of these values, but growth hormone, were within the normal range, whereas the level of growth hormone was slightly lower than normal range.
Although the GH level was low, comprehensive evaluations of hormones were not performed.
- subsequent pregnancies?
Response
Thank the reviewer for the important comment.
Because her cardiac function did not recover, she was considered to be at high risk for heart failure worsening during pregnancy. Therefore, she was not recommended to have any subsequent pregnancies. We added a comment about it in revised manuscript.
Change
Due to severely impaired cardiac function, she was not recommended to have any sub-sequent pregnancies.
- ICD, CRT indication?
Response
Thank the reviewer for the important comment. Her electrocardiogram showed narrow QRS complex, for which CRT was not recommended. In addition, there had been no episode of non-sustained ventricular tachycardia in electrocardiogram and Holter monitoring.
- COVID, autoimmun diseases, lues?
Response
Thank the reviewer for the important comment. Indeed, we experienced this case before covid pandemic era. There were no findings which might be associated with covid infection or covid virus vaccination. In addition, we measured autoantibody and MPO-ANCA, PR3-ANCA, which all exhibited no abnormalities. We added the comment about it in revised manuscript.
Change
We screened collagen disease, however the check of autoantibody and antineutrophil cytoplasm autoantibodies, which exhibited no abnormalities.
- Some minor mistakes in English expressions, e.g. coronary ischemia (ischemia is enough), etc.
Response
Thank the reviewer for the important comment. We corrected the parts in revised manuscript.

Reviewer 3 Report
Comments and Suggestions for Authors
The authors in this report described a case with co-occurrence of DCM and aortic dissection in which multi-vessel coronary artery dissection eventually occurred and consequently leading to advanced heart failure.
Several major points in this report should be addressed/clarified:
1) The chronological order of all events is not organized: From L38 till L108 the reader get a bit lost in the events since the authors describe the history in a reversed way. Authors should start their focus on main points of their manuscript and from there toward details and not vice versa.
2) In most case-reports all relevant events are summarized in a table. This is missing in the following paper.
3) How did the authors arrived to the initial diagnosis of postpartum DCM? Which guidelines criteria did they followed?
4) The authors stated that the patient did not achieve reverse remodeling in spite of optimal dosing of heart failure medications, and her cardiac function remained severely impaired. Infact , as the mentioned therapy is not an optimal medical therapy.
5) It was stated that the patient developed multi-vessel coronary artery dissection that re-exacerbated heart failure due to myocardial ischemia.
Infact, the LVEF was never normal, second the medical therapy was not optimized. Therefore, the contribution of myocardial ischemia is secondary.
6) Once talking about postpartum cardiomyopathy, then Bromocriptin is a main therapeutic option. It was not mentioned whether the patient was treated with bromocriptin.
Comments on the Quality of English LanguageModerate editing is needed
Author Response
Dear Editor and Reviewers
We thank the referee for fruitful suggestions. We have revised our manuscript on the basis of the referee’s comments. We look forward to a publication of our manuscript in your journal. Our responses to the referee’s comments are as follows:
1) The chronological order of all events is not organized: From L38 till L108 the reader get a bit lost in the events since the authors describe the history in a reversed way. Authors should start their focus on main points of their manuscript and from there toward details and not vice versa.
Response
Thank the reviewer for the important comment. In order to make the description more chronological, the diagnosis part and treatment part are described separately to avoid confusion. In addition, the following paragraph has been set for notes after discharge.
2) In most case-reports all relevant events are summarized in a table. This is missing in the following paper.
Response
Thank the reviewer for the important comment. In the revised manuscript, we presented all clinical events of hospitalization in Figure1.
3) How did the authors arrived to the initial diagnosis of postpartum DCM? Which guidelines criteria did they followed?
Response
Thank the reviewer for the pertinent comment.
As the statement from the European Society of Cardiology Study Group, peripartum cardiomyopathy is diagnosed as follows.
- Heart failure secondary to left ventricular systolic dysfuntion with a LVEF < 45%
- Occurrence towards the end of pregnancy or in the months following delivery (mostly in the month following delivery)
- No other identifiable cause of heart failure
We cited the report in revised manuscript.
4) The authors stated that the patient did not achieve reverse remodeling in spite of optimal dosing of heart failure medications, and her cardiac function remained severely impaired. Infact , as the mentioned therapy is not an optimal medical therapy.
Response
Thank the reviewer for the pertinent comment. As pointed out by the reviewer, it is not the case that oral medications have increased to the target dose. Furthermore, ARNI and SGLT2 inhibitors were not available in Japan at that time. Based on the above, it is reasonable to conclude that reverse remodeling could not be achieved with the maximum tolerable dose of cardioprotective drugs. We added a comment in the revised manuscript.
Change
Despite maximum tolerable dosing of heart failure medications
Sacubitril/valsartan and sodium glucose cotransporter 2 inhibitors were not available at that time.
5) It was stated that the patient developed multi-vessel coronary artery dissection that re-exacerbated heart failure due to myocardial ischemia.
Infact, the LVEF was never normal, second the medical therapy was not optimized. Therefore, the contribution of myocardial ischemia is secondary.
Response
Thank the reviewer for the important comment. As the reviewer pointed out, the main cause of heart failure worsening was the decreased systolic function of left ventricle. However, her heart failure status was continuously maintained for about 5 years despite severely impaired left ventricular function, therefore, we searched for new causes of worsening heart failure. We added a comment about it in revised manuscript.
Change
Of course, the main cause of worsening heart failure was severe systolic dysfunction, but it is expected that additional other factors may have added to the worsening of heart failure.
6) Once talking about postpartum cardiomyopathy, then Bromocriptin is a main therapeutic option. It was not mentioned whether the patient was treated with bromocriptin.
Response
As the reviewer pointed out, the administration of bromocriptin is one option for the treatment of postpartum cardiomyopathy. However, it was not used in the current case.

Reviewer 4 Report
Comments and Suggestions for Authors
The article entitled, “Multivessel coronary artery dissection in a patient with co-occurrence of aortic dissection and dilated cardiomyopathy in the postpartum period." is a good read. Authors have provided a case study with a patient who experienced the simultaneous occurrence of dilated cardiomyopathy (DCM) and aortic dissection. This case study highlights the development of multivessel coronary artery dissection, which ultimately resulted in the progression of severe heart failure. Following a period of stabilization in heart failure, the exacerbation of myocardial infarction resulting from coronary dissection further compromised the individual's cardiac function. This was evidenced by the deterioration of mitral regurgitation and a decline in right ventricular performance. The authors have noted that there is a potential occurrence of coronary artery dissection as a complication in patients who have both dilated cardiomyopathy (DCM) and aortopathy. This particular complication has the potential to significantly impact the clinical progression of advanced heart failure.
Altogether this is an important and timely article, this reviewer has certain suggestions that would help produce a more comprehensive overview of the topic:
Comments:
1, The English of manuscript can be polished (minor) and there are few typological errors.
2, Authors can add one paragraph for abbreviations.
3, Role of immune cells are also very important factor in coronary artery diseases, therefore I would suggest adding few citations to put comprehensive view of this topic (PMID: 36093172; PMID: 34043424; PMID: 32311026; PMID: 36337927; etc.).
4, At least one additional Figure (illustration) may be provided as to highlight the summary or prospect of this study.
5, Authors should provide limitations to their study.
Comments on the Quality of English LanguageMinor editing of English language required
Author Response
Dear Editor and Reviewers
We thank the referee for fruitful suggestions. We have revised our manuscript on the basis of the referee’s comments. We look forward to a publication of our manuscript in your journal. Our responses to the referee’s comments are as follows:
Reviewer 4
The article entitled, “Multivessel coronary artery dissection in a patient with co-occurrence of aortic dissection and dilated cardiomyopathy in the postpartum period." is a good read. Authors have provided a case study with a patient who experienced the simultaneous occurrence of dilated cardiomyopathy (DCM) and aortic dissection. This case study highlights the development of multivessel coronary artery dissection, which ultimately resulted in the progression of severe heart failure. Following a period of stabilization in heart failure, the exacerbation of myocardial infarction resulting from coronary dissection further compromised the individual's cardiac function. This was evidenced by the deterioration of mitral regurgitation and a decline in right ventricular performance. The authors have noted that there is a potential occurrence of coronary artery dissection as a complication in patients who have both dilated cardiomyopathy (DCM) and aortopathy. This particular complication has the potential to significantly impact the clinical progression of advanced heart failure.
Response
Thank the reviewer for the pertinent comment.
Comments:
1, The English of manuscript can be polished (minor) and there are few typological errors.
Response
Thank the reviewer for the important comment.
The manuscript had been already proofread by a native speaker, and we checked the details again. Additionally, we received proofreading by another native speaker again before the submission of revised manuscript.
2, Authors can add one paragraph for abbreviations.
Response
Thank the reviewer for the comment. However, it is rare for other submitted papers in this journal to present abbreviation separately, so to keep pace with this, we decided not to present them all together.
3, Role of immune cells are also very important factor in coronary artery diseases, therefore I would suggest adding few citations to put comprehensive view of this topic (PMID: 36093172; PMID: 34043424; PMID: 32311026; PMID: 36337927; etc.).
Response
Thank the reviewer for the important comment. Indeed, coronary artery disease is caused by various factors, and there have been several recent reports that it is related to immune responses.
Change
Coronary artery disease is caused by various factors, and among these, the contribution of immune response has recently been attracting attention. Although it is possible that immune reaction had some contribution on the coronary events in the current case, this is not fully established and future research is desired.
# Rurik JG, Aghajanian H, Epstein JA. Immune Cells and Immunotherapy for Cardiac Injury and Repair. Circ Res. 2021 May 28;128(11):1766-1779. doi: 10.1161/CIRCRESAHA.121.318005. Epub 2021 May 27. PMID: 34043424; PMCID: PMC8171813.
4, At least one additional Figure (illustration) may be provided as to highlight the summary or prospect of this study.
Response
Thank the reviewer for the important comment. We added a figure which described the association between vascular events and heart failure in this case in revised manuscript. (Figure 3)
5, Authors should provide limitations to their study.
Response
Thank the reviewer for the important comment. We added a comment about limitation in revised manuscript.
Change
As a limitation, it is thought that the various abnormalities in this case are related to some kind of genetic factor, however the genetic factor could not be eventually identified. Although the GH level was low, comprehensive evaluations of hormones were not performed. Furthermore, screening for genetic abnormalities in family members had not been done adequately.

Round 2
Reviewer 3 Report
Comments and Suggestions for Authors
The authors addressed my comments, paper get improved
Comments on the Quality of English Languageminor editing